An issue of concern: unique truncated ORF8 protein variants of SARS-CoV-2

Hassan Sk. Sarif 1
Kodakandla Vaishnavi 2
http://orcid.org/0000-0001-8246-0075 Redwan Elrashdy M. 3
Lundstrom Kenneth 4
Pal Choudhury Pabitra 5
http://orcid.org/0000-0002-3441-9673 Abd El-Aziz Tarek Mohamed 6
Takayama Kazuo 7
http://orcid.org/0000-0002-3313-4393 Kandimalla Ramesh 8
http://orcid.org/0000-0002-0021-2033 Lal Amos 9
Serrano-Aroca Ángel 10
http://orcid.org/0000-0001-5478-526X Azad Gajendra Kumar 11
http://orcid.org/0000-0002-9519-6338 Aljabali Alaa A.A. 12
Palù Giorgio 13
Chauhan Gaurav 14
http://orcid.org/0000-0003-4724-9463 Adadi Parise 15
Tambuwala Murtaza 16
http://orcid.org/0000-0001-8080-7960 Brufsky Adam M. 17
Baetas-da-Cruz Wagner 18
http://orcid.org/0000-0002-2557-7768 Barh Debmalya 19
http://orcid.org/0000-0002-4775-2280 Azevedo Vasco 20
http://orcid.org/0000-0002-9243-5444 Bazan Nikolas G. 21
http://orcid.org/0000-0002-8031-9454 Andrade Bruno Silva 22
http://orcid.org/0000-0002-0805-2003 Santana Silva Raner José 23
http://orcid.org/0000-0002-4037-5857 Uversky Vladimir N. 24 vuversky@usf.edu
1 Department of Mathematics, Pingla Thana Mahavidyalaya , Maligram , India
2 Department of Life sciences, Sophia College For Women, University of Mumbai , Mumbai , India
3 Faculty of Science, Department of Biological Science, King Abdulaziz University , Jeddah , Saudi Arabia
4 PanTherapeutics , Lutry , Switzerland
5 Applied Statistics Unit, Indian Statistical Institute , Kolkata , India
6 Department of Cellular and Integrative Physiology, University of Texas Health Science Center at San Antonio , San Antonio , TX, United States
7 Center for iPS Cell Research and Application (CiRA), Kyoto University , Kyoto , Japan
8 Applied Biology, CSIR-Indian Institute of Chemical Technology , Hyderabad , India
9 Division of Pulmonary and Critical Care Medicine, Mayo Clinic Rochester , Rochester , NY, United States
10 Biomaterials and Bioengineering Lab, Centro de Investigacion Traslacional San Alberto Magno, Universidad Catolica de Valencia San Vicente Martir , Valencia , Spain
11 Department of Zoology, Patna University , Patna , India
12 Department of Pharmaceutics and Pharmaceutical, Yarmouk University , Irbid , Jordan
13 Department of Molecular Medicine, University of Padova , Padova , Italy
14 School of Engineering and Sciences, Tecnologico de Monterrey , Monterrey , Mexico
15 Department of Food Science, University of Otago, University of Otago , Dunedin , New Zealand
16 School of Pharmacy and Pharmaceutical Science, Ulster University , Coleraine , UK
17 Department of Medicine, Division of Hematology/Oncology, University of Pittsburgh School of Medicine , Pittsburgh , PA, United States
18 Translational Laboratory in Molecular Physiology, Centre for Experimental Surgery, College of Medicine, Federal University of Rio de Janeiro , Rio de Janeiro , Brazil
19 Centre for Genomics and Applied Gene Technology, Institute of Integrative Omics and 46 Applied Biotechnology (IIOAB) , Nonakuri , India
20 Departamento de Genetica, Ecologia e Evolucao, Instituto de Ciencias Biologicas, Universidade Federal de Minas Gerais , Belo Horizonte , Brazil
21 Neuroscience Center of Excellence, School of Medicine, LSU Health New Orleans , New Orleans , LA, United States
22 Laboratório de Bioinformática e Química Computacional, Departamento de Ciências Biológicas, Universidade Estadual do Sudoeste da Bahia , Jequié , Brazil
23 Departamento de Ciencias Biologicas (DCB), Programa de Pos-Graduacao em Genetica e Biologia Molecular (PPGGBM), Universidade Estadual de Santa Cruz (UESC) , Ilheus , Brazil
24 Department of Molecular Medicine, University of South Florida , Tampa, FL , United States
Chen Jun
Electronic publication date: 2022 Mar 21
Publication date: 2022
Volume: 10
Electronic Location ID: e13136
Received 2021 Sep 16; Accepted 2022 Feb 27
Copyright: © 2022 Hassan et al.
Copyright year: 2022
Copyright holder: Hassan et al.
License: This is an open access article distributed under the terms of the Creative Commons Attribution License, which permits unrestricted use, distribution, reproduction and adaptation in any medium and for any purpose provided that it is properly attributed. For attribution, the original author(s), title, publication source (PeerJ) and either DOI or URL of the article must be cited.
License URL: https://creativecommons.org/licenses/by/4.0/

Keywords: ORF8, SARS-CoV-2, COVID-19, Truncated, Intrinsically disordered region, Truncation mutation, Continent distribution

Funding: The authors received no funding for this work.

==============================
Open reading frame 8 (ORF8) shows one of the highest levels of variability among accessory proteins in Severe Acute Respiratory Syndrome Coronavirus 2 (SARS-CoV-2), the causative agent of Coronavirus Disease 2019 (COVID-19). It was previously reported that the ORF8 protein inhibits the presentation of viral antigens by the major histocompatibility complex class I (MHC-I), which interacts with host factors involved in pulmonary inflammation. The ORF8 protein assists SARS-CoV-2 in evading immunity and plays a role in SARS-CoV-2 replication. Among many contributing mutations, Q27STOP, a mutation in the ORF8 protein, defines the B.1.1.7 lineage of SARS-CoV-2, engendering the second wave of COVID-19. In the present study, 47 unique truncated ORF8 proteins (T-ORF8) with the Q27STOP mutations were identified among 49,055 available B.1.1.7 SARS-CoV-2 sequences. The results show that only one of the 47 T-ORF8 variants spread to over 57 geo-locations in North America, and other continents, which include Africa, Asia, Europe and South America. Based on various quantitative features, such as amino acid homology, polar/non-polar sequence homology, Shannon entropy conservation, and other physicochemical properties of all specific 47 T-ORF8 protein variants, nine possible T-ORF8 unique variants were defined. The question as to whether T-ORF8 variants function similarly to the wild type ORF8 is yet to be investigated. A positive response to the question could exacerbate future COVID-19 waves, necessitating severe containment measures.

Introduction

The world is proceeding through a very difficult time due to the Coronavirus Disease 2019 (COVID-19), of which the causative agent is the Severe Acute Respiratory Syndrome Coronavirus 2 (SARS-CoV-2) (Hu et al., 2021; Yuen et al., 2020; Matheson & Lehner, 2020; Wu et al., 2020; Fontanet et al., 2021). There are nine open reading frames (ORFs), which encode accessory proteins important for the modulation of metabolism in infected host cells and innate immunity evasion via a complicated signalome and an interactome (Ren et al., 2020; Hassan et al., 2021c, 2020b; Díaz, 2020; Stukalov et al., 2021). The ORF8 protein is one of the most rapidly evolving accessory proteins among the beta coronaviruses, not only due to its ability to interfere with host immune response, but also with regards to its several missense mutations detected to date (Li et al., 2020; Zinzula, 2021; Hassan et al., 2021a; Flower et al., 2021). ORF8 directly interacts with major histocompatibility complex class I (MHC-I) both in vitro and in vivo, and is down-regulated, which impairs its ability to carry out antigen presentation and rendering infected cells less sensitive to lysis by cytotoxic T lymphocytes (Zhang et al., 2021). ORF8 suppresses type I interferon antiviral responses and interacts with host factors involved in pulmonary inflammation and fibrogenesis (Zhang et al., 2021; Rashid et al., 2021). From all viral proteomes interacting with human metalloproteome, the ORF8 proteins interact with 10 out of 58 human metalloproteins (Chasapis et al., 2021). Both SARS-CoV-2 and SARS-CoV ORF8 proteins play crucial roles in virus pathophysiological events, and dysregulate the TGF-β pathway, which is involved in the tissue fibrosis (Pereira, 2020). The functional implications of SARS-CoV-2 ORF8 had already gained immense attention and ORF8 is considered an important component of the immune evasion machinery (Li et al., 2020; Pereira, 2020; Mohammad et al., 2020; Alkhansa, Lakkis & El Zein, 2021). The SARS-CoV-2 ORF8 protein has less than twenty percent amino acid sequence homology with the SARS-CoV ORF8 protein, and represents a rapidly evolving viral protein (Flower et al., 2021; Velazquez-Salinas et al., 2020). A molecular framework for understanding the rapid evolution of ORF8, its contributions to COVID-19 pathogenesis, and the potential for its neutralization by antibodies have been supported by the structural analysis of the ORF8 protein (Hachim et al., 2020; Wang et al., 2020a). The crosstalk between SARS-CoV-2 or SARS-CoV infection and the host cell proteome at different levels may enable the identification of distinct and common molecular mechanisms (Zhang et al., 2021). Of note, SARS-CoV-2 ORF8 interacts with a significant number of host proteins related to the endoplasmic reticulum quality control, glycosylation, and extracellular matrix organization, although the mechanism of action of ORF8 concerning those interacting proteins is uncertain (Wang et al., 2020a; Wu et al., 2021).

The clade S, a subtype of SARS-CoV-2, was identified to possess the mutation L84S in the ORF8 protein sequence (Koyama, Platt & Parida, 2020; Mercatelli & Giorgi, 2020; Sengupta, Hassan & Choudhury, 2021). Presently, among many variants of SARS-CoV-2, the lineage B.1.1.7 carries a larger than usual number of genetic changes (Galloway et al., 2021; Ramírez et al., 2021; Frampton et al., 2021). Among many non-synonymous mutations, Q27STOP in the ORF8 protein contributed to the extrapolation of the branch leading to lineage B.1.1.7 (Perchetti et al., 2021; Li et al., 2021). The Q27STOP mutation inactivates the ORF8 protein favoring further downstream mutations, and could be responsible for the increased transmissibility of the B.1.1.7 variant (Galloway et al., 2021; Borges et al., 2021). The B.1.1.7 variant, being more transmissible than the wild-type SARS-CoV-2, was first detected in September 2020 in the UK (Shen et al., 2021; Walensky, Walke & Fauci, 2021). It began to spread rapidly by mid-December and was correlated with a significant increase in the number of SARS-CoV-2 infections in the UK and worldwide.

Functional implications on the immune surveillance due to the ORF8 truncation at position 27 remain unclear (Pereira, 2020). Therefore, it is of utmost importance to gain insight into the functionality of the truncated ORF8 protein variants to comprehend the B.1.1.7 lineage through theoretical and experimental characterization and genomic surveillance worldwide (Davies et al., 2021). The present study aimed at characterizing the unique variations of truncated ORF8 proteins (T-ORF8) due to the Q27STOP mutation. Further, this investigation differentiates a single T-ORF8 variant among 47 distinct unique T-ORF8 variants present in SARS-CoV-2 worldwide, as of May 20th, 2021. Several clusters of the unique T-ORF8 have been identified based on the various bioinformatics features and phylogenetic relationships, along with the emerging variants of the unique T-ORF8.

Data Acquisition and Methods

A total of 49,055 truncated ORF8 protein (T-ORF8) sequences (complete) from five continents (Asia, Africa, Europe, South America, and North America) were downloaded in FASTA format from the National Center for Biotechnology Information (NCBI) database (http://www.ncbi.nlm.nih.gov/). Note that no T-ORF8 protein sequence was found from Oceania. Next, FASTA files were processed in MATLAB 2021a for extracting unique T-ORF8 protein sequences for each continent. Note that only 47 unique T-ORF8 protein sequences were found. Here “unique T-ORF8” refers to a T-ORF8 protein sequence, which is different from other T-ORF8 sequences by the arrangement of amino acids, showing a non-zero Hamming distance from other T-ORF8 sequences. Consequently, the amino acid sequence of a “unique T-ORF8” variant is non-identical to other T-ORF8 sequences.

Derivation of polar/non-polar sequences and associated phylogeny

Every amino acid in a given T-ORF8 sequence was identified as polar (Q) or non-polar (P). Thus, every unique T-ORF8 became a binary sequence with two symbols P and Q. Then, the homology of these sequences was determined using the Clustal Omega web-suite (https://www.ebi.ac.uk/Tools/msa/clustalo/) and then associated with the nearest neighbor phylogenetic relationship among the unique T-ORF8 variants. Further, unique T-ORF8 variants with distinct binary polar/non-polar sequences were extracted (Hassan et al., 2020a; Broome & Hecht, 2000).

Frequency distribution of amino acids and phylogeny

The frequency of each amino acid present in a T-ORF8 sequence was determined using standard bioinformatics routine in Matlab-2021a. For each T-ORF8 protein, a twenty-dimensional frequency-vector considering the frequency of standard twenty amino acids can be obtained. Based on this frequency distribution of amino acids several consequences were drawn. The distance (Euclidean metric) between any two pairs of frequency vectors was calculated for each pair of T-ORF8 sequences. The distance matrix was used to develop a phylogenetic relationship based on the nearest neighbor-joining method using the standard routine in Matlab-2021a (Hassan et al., 2020c; Hassan, Choudhury & Roy, 2021).

Deep phylogenetic analyses

All phylogenetic data preparation and calculations were conducted in MEGA X (Kumar et al., 2018; Stecher, Tamura & Kumar, 2020). For deep phylogeny analyses we used two nucleotide datasets: one composed by the alignment of all 47 truncated ORF 8 sequences in addition to the RATG13 ORF8 as an external group, and a second group using all 47 truncated ORF 8, the RATG13 ORF 8 as well as 66 representative ORF 8 sequences for different Alpha, Beta, Gamma, Delta, Mu, GH/490R and Omicron variants. The phylogeny estimations were inferred by Maximum Likelihood using the Hasegawa-Kishino-Yano model (Hasegawa, Kishino & Yano, 1985). A discrete Gamma distribution was used compute the evolutionary rate differences among sites: four categories +G, parameter = 200.0000 and 500 bootstrap replications (Benvenuto et al., 2020). Furthermore, initial trees were heuristically obtained using Neighbor-Join and BioNJ algorithms with a Maximum Composite Likelihood (MCL) approach.

Amino acid conservation in terms of Shannon’s entropy

The degree of conservation of amino acids embedded in a T-ORF8 protein was obtained by the well-known information-theoretic measure called “Shannon’s entropy (SE)”. For each T-ORF8 protein, Shannon’s entropy of amino acid conservation over the amino acid sequence of T-ORF8 protein was calculated using as described in (Hassan et al., 2020c, 2021b). For a given T-ORF8 sequence of length l (here l = 26), the conservation of amino acids was calculated as follows:

SE=−∑i=120⁡psilog20(psi)

where psi=kil; ki represents the number of occurrences of an amino acid si in the T-ORF8 sequence (Strait & Dewey, 1996).

Prediction of molecular and physicochemical properties

Theoretical pI (PI), extinction coefficient (EC), instability index (II), aliphatic index (AI), protein solubility (PS), grand average of hydropathicity (GRAVY), and the number of tiny, small, aliphatic, aromatic, non- polar, polar, charged, basic and acidic residues of all unique T-ORF8 proteins were calculated using the web-servers ’ProtParam’, ’Protein-sol’ and EMBOSS Pepstats (Gasteiger et al., 2005; Hebditch et al., 2017; Madeira et al., 2019).

Intrinsic disorder analysis

All 47 T-ORF8 variants were subjected to the per-residue disorder analysis, for which PONDR® VSL2 algorithm was employed (Obradovic et al., 2005). This tool showed good performance on proteins containing both structure and disorder and was favorably ranked in a recent Critical Assessment of protein Intrinsic Disorder prediction (CAID) experiment (Necci, Piovesan & Tosatto, 2021).

Finding functional motifs

The Eukaryotic Linear Motif (ELM) resource (http://elm.eu.org/) was used for finding functional sites in proteins (Kumar et al., 2020). ELMs (also known as short linear motifs (SLiMs)), are short protein interaction sites, which are commonly found in intrinsically disordered regions of proteins and define a wide range of protein functionality.

Results

All 49,055 unique T-ORF8 protein variants were segregated from a set of available truncated ORF8 protein sequences collected from the NCBI database. Further, variability and commonality of the unique T-ORF8 proteins were analyzed from various quantitative measures as discussed in the methods section.

Characteristics of the unique variants of T-ORF8

The number of total sequences for each continent, the unique truncated ORF8 (T-ORF8) sequences, and their percentages are presented in Table 1. The results showed that 47 unique T-ORF8 proteins among the total of 48,691 were present in North America (Table 1). The unique T-ORF8 variants from Africa, Asia, Europe, and South America were contained in the set of unique T-ORF8 variants available in North America. Additionally, there were seven T-ORF8 with amino acid lengths 22, 24, 40 and 41 available in North America as of May 18, 2021 (Table 2). Note that among the seven T-ORF8 sequences, only five were found to be unique, as mentioned in Table 2. As of May 18, 2021 a single copy of the T-ORF8 proteins of amino acid lengths of 24 and 41 (Table 2) were found. There were two T-ORF8 variants of 41 amino acids available in North America. The most frequent T-ORF8 proteins so far observed, were the T-ORF8 proteins of 26 amino acids. It was observed that the T-ORF8 arose due to truncation at the residue positions 23, 25, 27, 41, and 42 of the 121 amino acid full-length ORF8 protein. We investigated the possible mutations for such truncations. A snapshot of the amino acid residues and their possible mutations with respect to the reference sequence NC 045512 is presented in Fig. S1.

Table 1 Frequency and percentages unique T-ORF8 variants (continent-wise).

Percentages of the unique T-ORF8 variants on continents	
Continent	Total T-ORF8 (T)	Unique T-ORF8 (U)	Percentage, continent-wise	Percentage, world-wide	
Africa	108	1	0.926%	1.96%	
Asia	99	1	1.01%	1.96%	
Europe	156	1	0.641%	1.96%	
South America	1	1	100%	1.96%	
North America	48,691	47	0.096%	92.16%	
World-wide	49,055	47	0.104%		
Note:

Here ‘U’ stands for the total number of unique T-ORF8 variants over the total available T-ORF8 sequences, which is denoted by ‘T’.

Table 2 Truncated ORF8 variants of length other than 26 amino acids.

Accession ID	Length (number of amino acid residues)	Date of collection	Geo-location	Remarks	
QQX22250.1	22	20-10-2020	USA: KS	2*Identical sequence	
QQX22346.1	22	24-09-2020	USA: MO		
QVF74147.1	24	27-04-2021	USA: Colorado	Worldwide frequency: 01	
QRE01295.1	40	13-12-2020	USA: MD	Worldwide frequency: 01	
QQX21038.1	41	30-10-2020	USA: OK	Worldwide frequency: 01	
QLJ58176.1	41	09-04-2020	USA	2*Identical sequence	
QLJ58236.1	41	16-04-2020	USA		

Note that at four positions, 23, 25, 27, and 40, amino acids glutamine (Gln) and cysteine (Cys) were truncated due to mutations at the codon’s first and third positions, respectively. The amino acid valine (Val) was truncated due to three mutations at the third, second, and first positions of the codon ’GUG’. Furthermore, it was observed that the mutations at the positions 23 and 25 were identical (C to U), and the changes of bases were transition mutations; i.e., pyrimidine (purine) to pyrimidine (purine). In contrast, the changes of bases of the truncated mutations at the positions 25 and 41 were transversal mutations; i.e., pyrimidine (purine) to purine (pyrimidine). For position 42, three sequences of mutations were hypothesized, taking place at first, second, and third positions of the codon (GUG); i.e., transition mutations (purine to purine), transversal mutation (pyrimidine to purine), and transversal mutation (purine to pyrimidine), respectively.

The list of unique T-ORF8 sequences of 26 amino acids with their representative accession IDs and sequences is presented in Table S1. Further, it was found that the unique T-ORF8 variants from Africa, Asia, Europe and South America were identical in relation to P15, as illustrated in Table S1. The date of sample collection, geo-location and accession ID of the first identified SARS-CoV-2 containing unique T-ORF8 variants are presented in Table S2.

Based on Table S2, the ORF8 protein sequence P15 was found in 48395 copies of the B.1.17 SARS-CoV-2 lineage in North America. Besides, the P15 variant with the Q27STOP mutation in the B.1.1.7 lineage was found in Africa, Asia, Europe and South America with a frequency of 108, 99, 156, and 1, respectively. None of the other 46 T-ORF8 unique variants was found on any continent, as of May 18, 2021. So, 46 unique T-ORF8 sequences were exclusively found in North America. Therefore, the P15 TORF8 variant is of particular interest for its uniqueness due to its apparent prevalence in most of the B.1.17 lineages of SARS-CoV-2 from North America and other continents.

In Europe, the P15 variant was first detected in two infected patients from Poland on March 15, 2020. In North America, two patients from Maryland were infected with the same SARS-CoV-2 P15 variant on May 27, 2020. After five days of the second occurrence of P15 in North America, one patient from Punjab-Pakistan (Asia) was infected by the P15 SARS-CoV-2 variant. Six months thereafter, the same variant was found in a patient from Ghana, for the first time in Africa. Twenty days after the fifth occurrence in Africa, the P15 variant was identified in Peru (South America) on December 31, 2020

Additionally, the frequency distribution of the T-ORF8 P15 variants across the North American continent is presented in Table 3. It was found that the T-ORF8 P15 variant spread over three geo-locations, Michigan, Florida, and Minnesota, with the highest number of frequencies of 5084, 6884, and 7416, respectively. The P15 variant was found for the first time in Maryland, but the frequency at this geo-location was 1171 on May 14, 2021.

Table 3 Distribution of cumulative frequency of P15 variants across North America.

Geo-location	Frequency	Geo-location	Frequency	Geo-location	Frequency	
Wyoming	56	North Carolina	776	Iowa	141	
Wisconsin	383	New York	887	Indiana	823	
West Virginia	289	New Mexico	250	Illinois	1,426	
Washington	83	New Jersey	1,815	Idaho	85	
Virginia	917	New Hampshire	234	Hawaii	16	
Vermont	209	Nevada	157	Guam	7	
Utah	97	Nebraska	105	Georgia	1,232	
Texas	3,420	Montana	25	Florida	6,884	
Tennessee	993	Missouri	254	District of Columbia	61	
South Dakota	86	Mississippi	40	Delaware	70	
South Carolina	261	Minnesota	7,416	Connecticut	496	
Rhode Island	339	Michigan	5,084	Colorado	533	
Puerto Rico	224	Massachusetts	2,761	California	1,727	
Pennsylvania	3,285	Maryland	1,171	CA, Santa Clara County	4	
Oregon	166	Maine	79	CA, Humboldt	20	
Okhlahoma	81	Louisiana	223	Arkansas	62	
Ohio	1,191	Kentucky	145	Arizona	290	
North Dakota	13	Kansas	100	Alaska	65	
				Alabama	168	
				USA	654	

There were 18 geo-locations, where the frequency of spread of the P15 variant was found to be less than 100 (Table 3). Among other geo-locations, Guam (US territory located in the Pacific Ocean) and North Dakota, USA, had the least number of patients infected by the B.1.1.7 variant containing the P15 protein. In Guam, according to the NCBI SARS-CoV-2 database, all seven patients were infected by the B.1.1.7 variant containing the P15, within a short period from February 21 to April 11, 2021. Also, in North Dakota, 13 of 16 patients were infected by the same strain of SARS-CoV-2 from February 2, 2021 to April 28, 2021.

The frequency distribution of all T-ORF8 variants across the US is presented in Table 4. It is evident that all 47 unique T-ORF8 variants were detected in 21 different states of the US. The highest number (7) of the unique T-ORF8 variants was detected as a first instance in Pennsylvania within a short period (March 2 to April 20, 2021) (Table 4). The P35 variant was found initially in two states: Rhode Island and Massachusetts on March 20, 2021. Furthermore, it was observed that all T-ORF8 variants other than P15 emerged for the first time from February 12, 2021 to April 28, 2021.

Table 4 Frequency distribution of unique T-ORF8 variants over the USA.

USA: states	Unique T-ORF8 variants	USA: states	Unique T-ORF8 variants	
USA: California	P1, P30, P40	USA: Missouri	P28	
USA: Connecticut	P32, P33	USA: New Jersey	P10, P41	
USA: Florida	P4, P14, P16	USA: Ohio	P2	
USA: Georgia	P21	USA: North Carolina	P47	
USA: Illinois	P18	USA: Pennsylvania	P7, P17, P19, P20, P23, P27, P39	
USA: Kentucky	P36	USA: Puerto Rico	P24	
USA: Louisiana	P31	USA: Tennessee	P5, P22	
USA: Maryland	P15, P26, P29, P34	USA: Rhode Island	P35	
USA: Massachusetts	P35, P42	USA: Texas	P6, P9, P11, P25	
USA: Michigan	P8, P38, P43, P46	USA: Utah	P45	
USA: Minnesota	P3, P12, P13, P37, P44			

Using the Clustal Omega web-server, an amino acid sequence-based alignment and corresponding phylogenetic tree of the unique T-ORF8 variants are presented in Fig. S2. From the sequence alignment, it was derived that all unique T-ORF8 variants share identical amino acids methionine (Met), lysine (Lys), Gln, serine (Ser), and leucine (Leu) at positions 1, 2, 18, 21, and 22 respectively. Further it was found that T-ORF8 P15 is closest to the ORF8 sequences P13 and P14 (Fig. S2). Note that P15 was placed at the leftmost branch of the phylogenetic tree, which made the sequence P15 distinguishable from the rest of the T-ORF8 variants. The pairs of T-ORF8 variants (P13, P14), (P5, P6), (P33, P45), (P9, P37), (P19, P20), (P35, P36), (P31, P34), (P29, P43), (P27, P42), (P25, P26), (P22, P23), (P21, P40), (P17, P18), (P2, P8), and (P1, P47) were found to be the closest to each other based on the amino acid sequence homology-based phylogeny (Fig. S2).

Next, we conducted deep phylogenic analysis. The first nucleotide dataset used for this analysis included the alignment of all 47 truncated ORF8 sequences in addition to the RATG13 ORF8 as an external group. Phylogenic analysis of this group returned a tree (Fig. 1) with the highest log likelihood of −400.62 in an analysis with 48 nucleotide sequences and a total of 83 positions in the final dataset. In addition, Fig. 1 shows nine well-defined monophyletic groups, where the ORF 8 sequences 15 and 40 formed a unique clade in the group II. Based on this tree analysis it can be inferred that at least nine truncated ORF8 variants could be related to a specific SARS-CoV-2 variant and/or a specific variant subtype, in this case more related to de Alpha type. Interestingly, sequences 35 and 46 rooted outside the tree together with the external group ORF8 from the RATG13 genome. In these cases, the sequences 35 and 46 are possibly mutating to the corresponding SARS-CoV-2 start point or they could be related to a subtype with a small viral load and spread.

Figure 1 Maximum likelihood phylogenetic tree for the 47 truncated ORF8, using 500 bootstrap replications and the Hasegawa-Kishino-Yano model.

Nine group clades were found, while sequences 35 and 46 (marine blue and purple arrows, respectively) are phylogenetically near to the RATG13 ORF8 sequence. Sequence 15 is indicated by a red arrow.

In the second dataset there were all 47 truncated ORF8 variants, the RATG13 ORF8, as well as 66 representative ORF8 sequences from different Alpha, Beta, Gamma, Delta, Mu, GH/490R, and Omicron variants. Results of the phylogenetic analysis of this set containing all 47 sequences and different SARS-CoV-2 ORF8 variant representatives are shown in Fig. S3, where the resultant tree presented a highest log likelihood of −430.29.

This analysis involved 114 nucleotide sequences with a total of 81 positions in the final dataset. In this tree, one can find at least 14 group clades, and, as expected, almost all ORF8 variants are grouped together in the group I. The other 47 truncated ORF8 sequences presented similar tree topologies in comparison to the first dataset tree (cf. Figs. 1 and S3). On the other hand, the group IV showed that the sequences 15 and 40 formed a clear clade to the Alpha ORF8, what was expected. Furthermore, these sequences grouped with the ORF8 variant B.1.640, which was described as the new variant IHU found in France in December 2021 (Colson et al., 2021).

Evaluation of intrinsic disorder content of 47 T-ORF8 proteins

Unfortunately, no structural information is available for the truncated forms of ORF8. In fact, although two X-ray structures were reported for the dimeric form of the SARS-CoV-2 ORF8 protein, both of these structures were solved for the ORF8 sequences containing residues 18–121; i.e., they do not include information for more than a half (63%) of the N-terminal region of the full-length ORF8, which constitutes T-ORF8. Furthermore, truncated forms of ORF8 are relatively short to have independently foldable structure. It would be really important to conduct some structural analysis of the truncated ORF8 (e.g., by solution NMR). However, such an analysis could be complicated by the presence of two cysteine residues and by the highly hydrophobic nature of this region showing the mean hydrophobicity of 0.5926, which dramatically exceeds the mean hydrophobicity of typical globular proteins (~0.46 ± 0.05).

Therefore, to gain some structure-related information for the truncated forms of ORF8, we analyzed the peculiarities of the distribution of per-residue intrinsic disorder predisposition within sequences of the 47 T-ORF8 variants. Since the amino acid sequences of T-ORF8 proteins are shorter than 30 residues, the number of computational tools capable of predicting the intrinsic disorder is limited. In this study, we used the PONDR® VSL2 algorithm. The results of this analysis are shown in Fig. 2. Due to their short length and limited sequence variability, T-ORF8 proteins are characterized by rather featureless disorder profiles, where both N- and C-terminal regions are predicted to have higher levels of intrinsic disorder than the central parts.

Figure 2 Analysis of intrinsic disorder predisposition of 47 T-ORF8 proteins.

(A) Disorder profiles generated using the PONDR-VSL2 disorder predictor. Three thresholds of predicted disorder scores (PDSs) are shown, 0.15, 0.25, and 0.5, which are used for the classification of protein residues as highly disordered (PDS ≥ 0.5), flexible (0.25 ≤ PDS < 0.5), moderately flexible (0.15 ≤ PDS < 0.25) and mostly ordered (PDS < 0.15). B. Ranking of 47 T-ORF8 proteins based on their mean disorder scores.

Most T-ORF8 proteins show rather similar profiles, with the noticeable exceptions of P1 and P36, which show the highest disorder levels in their N-terminal regions. In contrast, P45 presents the least disorder at the N-terminus, P25 the longest and most peculiar disorder distribution in its C-terminal half, P18 and P19 long disorder stretches at their C-termini, and P12 the lowest levels of disorder in the C-terminal region (Fig. 2A). These observations are further supported by Fig. 2B, where the 47 T-ORF8 proteins are ranked based on their mean disorder scores, from the highest to the lowest levels of disorder. However, the vast majority of T-ORF8 proteins (38 of 47) form a rather uniform cluster with the average mean disorder score of 0.304 ± 0.010, whereas P25, P19, P36, P18, and P1 show higher than average and P13, P40, P23, and P12 lower than average levels of disorder.

Table S3 lists potential functional motifs identified in the 47 T-ORF8 variants by ELM resource and shows that all T-ORF8 proteins have several such motifs. Based on their content of functional motifs, T-ORF8 proteins can be grouped into 21 clusters, with three clusters containing 13, 4, and 2 proteins, respectively, and all the remaining being singletons. The common motif found in all T-ORF8 proteins is the N-degron that initiates protein degradation by binding to the UBR-box of N-recognins. A kinase docking motif that mediates interaction towards the ERK1/2 and P38 subfamilies of MAP kinases and a Ser/Thr (serine/threonine) residue phosphorylated by the Plk1 kinase are present in 20 clusters, whereas 17 clusters also include a site for attachment of a fucose residue to a serine. The lowest number of functional motifs (3) is found in 6 proteins (P12, P16, P17, P19, P21, and P40), many of which are characterized by lower mean disorder scores. On the contrary, proteins with the largest number of functional motifs (6 and 7) typically show higher disorder scores. Table S3 shows that truncation might generate functional T-ORF8 variants (or at least variants possessing functional motifs), and that expected functionality of different T-ORF8 proteins can be quite different. It is clear that the results of this computational analysis should be taken with caution, and the functionality of T-ORF8 requires experimental validation.

Variability and commonality of T-ORF8 variants

In the proceeding section, the quantification of unique T-ORF8 variants using various parameters such as polar/non-polar residue sequence homology, amino acid frequency distributions, amino acid conservation through the Shannon entropy, and physicochemical properties is described.

Polarity based variability of T-ORF8 variants

Each unique T-ORF8 variant possessed a binary polar/non-polar sequence and based on the sequence homology of these sequences, a phylogenetic relationship was obtained (Fig. S4). The number of polar and non-polar residues in the unique T-ORF8 variants was found to be almost balanced (50-50 in percentage). Among 26 residue positions of each T-ORF8 variant with the amino acid length of 26 residues, 14 positions (polar residues at positions 1, 5–7, 13 and non-polar residues at positions 2, 17–18, 20–23) remained invariant as illustrated by Table S4. The pairs of unique T-ORF8 variants (P5, P6), (P21, P40), (P4, P33), (P12, P13), (P28, P29), and (P9, P10) were closest to each other (Fig. S4).

Note that, the P15 variant was placed in a single leaf and found to be distant from the other unique ORF8 variants as per polarity-based homology, although P15 was closest to the T-ORF8 variants P13 and P14 based on amino acid homology. Furthermore, it was noticed that only 17 unique T-ORF8 variants possessed unique polar/non-polar sequences (Table S4). The polar/non-polar sequence of each T-ORF8 variant other than P4, P5, P15, and P28 was unique.

Surprisingly, among the total of 47 T-ORF8 variants, there were 28 T-ORF8 variants, which share identical polar/non-polar sequence with that of P15. According to the phylogenetic relationship derived from the unique polar/non-polar sequence homology, the T-ORF8 P15 was found to be closest to P42. Furthermore, the pairs (P13, P25), (P21, P40), and (P5, P30) were found to be close enough to each other (see Fig. S5).

Variability of the frequency distribution of amino acids present in T-ORF8 variants

The frequency of each amino acid present in the unique T-ORF8 variants was enumerated, and consequently, a twenty-dimensional frequency vector was obtained (Table S5). Tryptophan (Trp) was not present in any of the unique T-ORF8 variants. It was noted that the amino acids arginine (Arg), asparagine (Asn), aspartic acid (Asp), proline (Pro), and tyrosine (Tyr) were absent in the T-ORF8 P15. Arg was found with frequency one in the T-ORF8 P17, P39, P19, P22, and P33. In the P14 and P41 sequences, Asn was present with frequency one. Likewise, Asp was found in P38 and P11. Pro with frequency one, was found in the P8 variant only. Tyr was found in the T-ORF8 P20 and P23 variants. The highest frequency of 4 was seen for phenylalanine (Phe) in P3, P16, and P47, Leu in P10, P12, P24, P34, and P35, and threonine (Thr) in P29, P31, P32, and P42.

For each pair of frequency vectors corresponding to all unique T-ORF8 variants, Euclidean distances were calculated (Table S6), and the distance matrix in color heat-map is presented in Fig. 3. It was found that the P15 variant is equidistant (1.41) from all other variants except P30 and P40, which are one distance apart from P15. Further, we observed that the distance between any two pairs of T-ORF8 variants is two (light green color) except in a few cases (Fig. S6). Although the amino acid sequences were different, identical frequency vectors were found for the pair of ORF8 variants (P3, P47), (P2, P9), (P6, P13), (P17, P19), (P24, P34), (P24, P35), (P25, P36), (P34, P35), (P29, P42), and (P27, P43).

Figure 3 Pairwise distance matrix of amino acid frequency vectors of the unique T-ORF8 variants.

Based on the distance matrix, all unique T-ORF8 variants were clustered, and the associated phylogeny is presented in Fig. S6. The P15 variant was very close to P22, P23, P33, P40, and P41 according to the phylogenetic relationship depicted in Fig. S6. Other than the pairs of T-ORF8 having identical frequency vectors, it was found that the pairs of the unique T-ORF8 variants (P23, P33), (P14, P30), (P19, P20), and (P31, P32) were close to each other as derived from the phylogenetic relationship (Fig. S6).

Variability of T-ORF8 through Shannon entropy

Shannon entropy (SE) for each unique T-ORF8 variant was calculated using the formula stated in “Deep Phylogenetic Analyses” (Table S7). It was found that the highest and lowest SEs of the 47 unique T-ORF8 proteins were 0.958 and 0.973, respectively. That is, the length of the largest interval is 0.015, which is sufficiently small. Based on SEs of the T-ORF8 proteins a set of clusters were derived (Fig. S7A) SEs of each of the T-ORF8 variants are plotted in Fig. S7B. The largest cluster containing 18 T-ORF8 variants (including the T-ORF8 P15) based on the identical SEs were obtained (Fig. S7).

Molecular and physicochemical informatics of unique T-ORF8 variants

For each unique T-ORF8 variant and complete ORF8 protein, several physicochemical and molecular properties were computed using the web-servers. It was found that the extinction coefficient of all T-ORF8 variants was 125, except for four T-ORF8 variants, P16, P17, P18, and P19, whose extinction coefficient was zero (see Table S8). Further, it was noticed that for P20 and P23, extinction coefficients were found to be significantly higher compared to the others. Instability indices of all the T-ORF8 protein variants were ranging from 45.36 to 95.85 (greater than 40), and consequently they are all unstable. It was observed that the P15 variant had a unique frequency of the various types of residues (Tiny: 10, Small: 12, Aliphatic: 9, Aromatic: 4, Non-polar: 16, Polar: 10, Charged: 3, Basic: 2, Acidic: 1) and none of the other T-ORF8 variants was identical.

Furthermore, Euclidean distances between every pair of molecular and physicochemical property vectors corresponding to each T-ORF8 variant were computed and a phylogenetic relationship was derived (Fig. 4) based on the distance matrix (Table S9). Note that the property vectors of P20 and P30 were highly distant from the other ORF8 variants due to the huge difference in extinction coefficients (for P20, EC: 1490 and for P30, EC: 1615). So, ignoring these two ORF8 variants, the phylogenetic relationship among the remaining 45 T-ORF8 was derived. It was found that none of the T-ORF8 variants had identical property vectors to the P15 variant. It was further found from the phylogenetic relationship (Fig. 4B) that the pair of unique T-ORF8 variants (P17, P18), (P8, P22), (P4, P33), (P28, P29), (P2, P9), (P27, P43), (P7, P39), (P15, P30), (P25, P36), (P26, P45), (P24, P34), (P11, P14), (P21, P40), (P3, P47), and (P31, P32) were found to be the closest pairs based on the closeness of property vectors. Property vector distances from each 45 unique T-ORF8 variant from the P15 variant are presented in Table S10. In the close vicinity of P15, only the P25, P30, and P36 variants appeared based on the nearness of property vectors (Table S10).

Figure 4 Distance matrix of property vectors and derived phylogenetic tree of 45 T-ORF8 variants.

(A) The distance matrix; (B) phylogenetic tree based on physicochemical properties.

Possible T-ORF8 variants in the likelihood of P15 variant

Based on the amino acid sequence homology and other various features, such as the frequency distribution of amino acids, SE, and physicochemical properties of T-ORF8 variants, a possible cluster of nine unique T-ORF8 variants are derived. A schematic presentation is given in Fig. 5. All these nine unique T-ORF8 variants had unique polar/non-polar sequences as shown in Table S4. In addition to the P15 variant, these possible nine emerging variants are likely to appear in the B.1.1.7 lineage of SARS-CoV-2 in the near future. As of May 22nd, 2021, it was observed that 16 of 17 COVID-19 affected patients from India (mostly from Gujrat), were infected by the B.1.1.7 lineage of SARS-CoV-2 with the P15 variant, and only one patient (Accession: QVO43928) infected on February 28, 2021 with the SARS-CoV-2 strain with the P34 T-ORF8 variant, which had an identical polar/non-polar sequence as that of P15.

Figure 5 A schematic representation of a possible cluster of unique T-ORF8 variants which were residing in the likelihood of P15 variant.

Note: the frequency of each length of T-ORF8 protein was mentioned in parentheses. T-ORF8 variants mentioned in each box were found in the close likelihood of P15 T-ORF8 variants.

Discussion and Concluding Remarks

The ORF8 protein is 121 amino acid long with two genotypes (orf8L and orf8S). It has an Ig-like fold, is highly immunogenic, and interacts with 47 human proteins, 15 of them being drug targets interacting with MHC-I molecules leading to a significant down-regulation of their surface expression on various cell types (Rashid et al., 2021; Gordon et al., 2020; Gamage et al., 2020). As a result, inhibition of ORF8 could boost special immune surveillance and speed up SARS-CoV-2 eradication in vivo (Gamage et al., 2020). ORF8 is not like ORF3a, an ion channel (viroporin), implicated in virion assembly and membrane budding both in SARS-CoV and SARS-CoV-2. In contrast to SARS-CoV, which is not viable when lacking E and ORF3a proteins, and which requires the full-length E and ORF3a proteins for maximal replication and virulence (Barrantes, 2021; Castaño-Rodriguez et al., 2018; Tan et al., 2021), ORF8 in SASR-CoV-2 seems to only have a minor or no impact on the viral life cycle, as the virus can seemingly survive without a functional ORF8 protein, which has been demonstrated by the presence of many mutations and truncations detected in viable SARS-CoV-2 variants (Wang et al., 2020a, 2021). The Q27STOP mutation in the ORF8 protein has been discovered to cause 47 distinct truncated ORF8 variations. Furthermore, other truncated protein variants of different lengths 22, 24, 40, and 41 amino acids, were detected, although the frequency of occurrences of those variants was significantly lower (Table 2). In Colorado, one T-ORF8 variant of the length of 24 amino acids was detected on April 24, 2021, and this variant is likely to spread further in the future. Other truncated ORF8 variants of amino acids lengths of 22, 40, and 41, no longer appeared in new strains of SARS-CoV-2.

An important consequence of the ORF8 truncation is the alterations in the presentation of this protein by the human leukocyte antigen (HLA) complex on the surface of T-cells, which are among the key players in the immune response to viral infection. In fact, the development of the T-cell-based immunity is based on the presentation of short viral peptides on the cell surface by the HLA complex. Activation of killer CD8 T-cells depends on the recognition of viral peptides presented by HLA class I molecules, whereas activation of CD4 T lymphocytes (helper T-cells) is initiated by binding to a complex between viral peptides and HLA class II molecules (Klein & Sato, 2000). Although the major functions of the activated CD8 T-cells are the recognition and elimination of the infected cells, the main function of activated CD4 T-cells is regulation of immunity, including stimulation of antibody generation by B cells and enhancement of CD8 T-cell responses (Klein & Sato, 2000; Swain, McKinstry & Strutt, 2012). The overall importance of the T-cell-based responses in COVID-19 severity and long-term immunity has been documented in multiple reports (Wang et al., 2020b; Shkurnikov et al., 2021; Iturrieta-Zuazo et al., 2020; Bange et al., 2021; Grifoni et al., 2020; Peng et al., 2020; Dan et al., 2021). Therefore, it is not surprising that constant appearance of novel mutations in the SARS-CoV-2 genome can target T-cell epitopes (Agerer et al., 2021; Reynolds et al., 2021). Potential CD8 and CD4 epitopes represent peptides derived from the viral proteins, where 8- to 14-mers and 15- to 20-mers are serving as partners for HLA class I and class II receptors, respectively. The appearance of the premature stop codon in the ORF8 gene encoding the NS8 (ORF8) protein (Q27stop) not only generates the truncated form of this protein due to the deletion of almost 80% of the ORF8 protein, but also eliminates the whole immunopeptidome associated with this part of protein. Therefore, it is likely that such distortion in the immunopeptidome of ORF8 will dramatically affect its immunogenicity. Furthermore, mutations found in T-ORF8 could further modulate T-cell immunity.

Quantitative characteristics of the 47 unique truncated ORF8 protein variants were examined. All 47 T-ORF8 variants were found in North America, and only the P15 T-ORF8 variant was spread to four other continents: Africa, Asia, Europe and South America, until May 22, 2021. In this regard, it is pertinent to raise the question of whether there is any correlation between the spreading of all unique T-ORF8 variants and the epidemiological nature of North America. Within North America, it was reported that one of the top mutations, 27964C>T-(S24L) in the ORF8 protein, has an unusually strong gender dependence (Wang et al., 2021). The spread of the P15 variants over the 57 geo-locations across North America was noticed, and in addition, many patients from Asia, Africa, South America and Europe were infected by the particular B.1.1.7 variant of SARS-CoV-2, which contains the P15 variant. Like in many states of the US, also in the US territory of Guam and in North Dakota, most of the patients were noticed to have contracted the P15 variant of the B.1.1.7 lineage. Consequently, the present trend implies that a much higher spread of this lineage with this particular P15 variant is likely to occur. After Europe, the first case of the B.1.1.7 variant with the T-ORF8 P15 was discovered in Maryland in the US, but although later this strain remained limited in Maryland it spread further to other states, such as Florida and Minnesota (Table 3). Furthermore, this analysis reports a set of nine most likely T-ORF8 variants P4, P5, P13, P21, P25, P30, P36, P40, and P42, which were found to be residing in close vicinity of the P15 ORF8 variant. It was noticed that among 47 unique T-ORF8 variants, 28 of them had identical polar/non-polar sequences to that of the P15 variant. Considering the ability of the P15 variant to spread one can assume that the 28 variants with identical polar/non-polar sequences may spread in the near future and cause third, fourth, and fifth waves of COVID-19. As evidence, one patient from India was infected with SARS-CoV-2 with the P34 variant, which has the same polar/non-polar sequences as the P15 variant, as of May 22, 2021 (NCBI accession: QVO43928). The fact that T-ORF8 is still operating as ORF8, is an open issue that needs to be addressed. Reports try to link these T-ORF8 present in many lineages to COVID-19 severity and/or outcomes, effects that contribute to disease progression if associated with mutations in spike protein (Pereira, 2020; Guthmiller et al., 2021; Nagy, Pongor & Győrffy, 2021). It has also been reported that patients infected with SARS-CoV-2, lacking the majority of ORF8 protein, have a lower risk of aggravation, a conclusion that accrued variants in the spike, ORF8, and ORF3a proteins were associated with improved clinical outcomes (Esper et al., 2021; Young et al., 2020). More recently, SARS-CoV-2 strains were isolated in Washington state (USA), with a stop codon mutation generating a novel truncated and much shorter ORF8 protein, as well as in Hong Kong, which completely lacked ORF8 (gene, protein and antibody), ORF7a, and ORF7b (Esper et al., 2021; DeRonde et al., 2021; Tse et al., 2021). However, the in vitro analysis on Nasal Epithelial cells (NECs) infected with one of these isolates (ORF8-∆382) showed no significant functional differences between the wild type ORF8 and the ORF-∆382 mutant (Gamage et al., 2020). In contrast, Vero-6 cell infected with the same strain (ORF8-∆382) showed significantly higher replicative fitness in vitro than the wild type, while no difference was observed in the patient viral load, indicating that the deletion variant retained its replicative fitness (Su et al., 2020). In any case, the combinatorial clinical effects of T-ORF8 need to be investigated and analyzed in depth. It is necessary to investigate in detail the functions of T-ORF8 and the effects of this protein on inflammation and antigen-presenting ability. Finally, caution should be paid to using ORF8 as a diagnostic marker, as many immunoassay tests depend on antibody potency (Pereira, 2021). A systematic analysis of this protein and its specific antibodies is needed to determine the effects of these mutations/truncations on the diagnostic potential of the anti-ORF8 antibodies.

Supplemental Information

Supplemental Information 1 Supplementary figures and tables.

Click here for additional data file.

Additional Information and Declarations

Competing Interests

Author Contributions

Data Availability

Vasco Azevedo, Debmalya Barh and Vladimir N Uversky are Academic Editors for PeerJ.

Kenneth Lundstrom is employed by PanTherapeutics.

Sk. Sarif Hassan conceived and designed the experiments, performed the experiments, analyzed the data, prepared figures and/or tables, authored or reviewed drafts of the paper, and approved the final draft.

Vaishnavi Kodakandla performed the experiments, analyzed the data, authored or reviewed drafts of the paper, and approved the final draft.

Elrashdy M. Redwan performed the experiments, analyzed the data, authored or reviewed drafts of the paper, and approved the final draft.

Kenneth Lundstrom conceived and designed the experiments, performed the experiments, analyzed the data, authored or reviewed drafts of the paper, and approved the final draft.

Pabitra Pal Choudhury performed the experiments, analyzed the data, authored or reviewed drafts of the paper, and approved the final draft.

Tarek Mohamed Abd El-Aziz performed the experiments, analyzed the data, authored or reviewed drafts of the paper, and approved the final draft.

Kazuo Takayama performed the experiments, analyzed the data, authored or reviewed drafts of the paper, and approved the final draft.

Ramesh Kandimalla performed the experiments, analyzed the data, authored or reviewed drafts of the paper, and approved the final draft.

Amos Lal performed the experiments, analyzed the data, authored or reviewed drafts of the paper, and approved the final draft.

Ángel Serrano-Aroca performed the experiments, analyzed the data, authored or reviewed drafts of the paper, and approved the final draft.

Gajendra Kumar Azad performed the experiments, analyzed the data, authored or reviewed drafts of the paper, and approved the final draft.

Alaa AA. Aljabali performed the experiments, analyzed the data, authored or reviewed drafts of the paper, and approved the final draft.

Giorgio Palù performed the experiments, analyzed the data, authored or reviewed drafts of the paper, and approved the final draft.

Gaurav Chauhan performed the experiments, analyzed the data, authored or reviewed drafts of the paper, and approved the final draft.

Parise Adadi performed the experiments, analyzed the data, authored or reviewed drafts of the paper, and approved the final draft.

Murtaza Tambuwala performed the experiments, analyzed the data, authored or reviewed drafts of the paper, and approved the final draft.

Adam M. Brufsky performed the experiments, analyzed the data, authored or reviewed drafts of the paper, and approved the final draft.

Wagner Baetas-da-Cruz performed the experiments, analyzed the data, authored or reviewed drafts of the paper, and approved the final draft.

Debmalya Barh performed the experiments, analyzed the data, authored or reviewed drafts of the paper, and approved the final draft.

Vasco Azevedo performed the experiments, analyzed the data, authored or reviewed drafts of the paper, and approved the final draft.

Nikolas G. Bazan performed the experiments, analyzed the data, authored or reviewed drafts of the paper, and approved the final draft.

Bruno Silva Andrade conceived and designed the experiments, performed the experiments, analyzed the data, prepared figures and/or tables, authored or reviewed drafts of the paper, and approved the final draft.

Raner José Santana Silva performed the experiments, analyzed the data, prepared figures and/or tables, authored or reviewed drafts of the paper, and approved the final draft.

Vladimir N. Uversky conceived and designed the experiments, performed the experiments, analyzed the data, prepared figures and/or tables, authored or reviewed drafts of the paper, and approved the final draft.

The following information was supplied regarding data availability:

The data are available in the article and the Supplemental File.

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
