# Peer review of "An issue of concern: unique truncated ORF8 protein variants of SARS-CoV-2"

_PeerJ, doi:10.7717/peerj.13136_

## Round 0.1 · original submission · Major Revisions

Although three reviewers found your study interesting and analysis thorough, reviewer 2 raised significant concerns about your work. Please revise according to reviewer 2's comments.

·

Basic reporting

Long sentences in the discussion part. Spelling mistakes of ORF8 in the discussion.

Experimental design

Authors should provide detail about identification of T-ORF8 and matlab programs.

Validity of the findings

Discussion needs to be modified with more clarity.

·

Basic reporting

No comment

Experimental design

No comments

Validity of the findings

No comments

Additional comments

The hypothesis and conclusions were clearly identified and the importance of the study was stated making it easier to follow the authors' experimental design. Well laid out schematics and tables were also appreciated.

Reviewer 3 ·

Basic reporting

The manuscript writing and data organization is poorly organized. The concept of unique T-ORF8s has to be better explained.

Experimental design

The analysis of variants within USA is not relevant, and other factors may be analyses together with geography. Manuscript organization is confusing with unnecessary tables and figures that distract the reading.

Validity of the findings

The work requires a deep phylogenetic analysis to support major claims, which is lacking.

Reviewer 4 ·

Basic reporting

The paper reports the role of mutation in ORF8 of SARS CoV2 leads to second wave in several countries. the authors have done significant in silico analysis of orf8 from different regions.

Experimental design

experimental designing is very well planned. but there is a small suggestion if anything related to structure can be included it will be good and further enrich the manuscript.

Validity of the findings

the findings requires a experimental validation. but the in silico analysis has significant impact.

Additional comments

no comments

---

## Round 0.2 · accepted · Accept

Your manuscript is now suitable for publication. Thanks for the nice work